# Simulating Interclonal Interactions in Diffuse Large B-Cell Lymphoma

**DOI:** 10.3390/bioengineering10121360

**Published:** 2023-11-27

**Authors:** Siddarth R. Ganesh, Charles M. Roth, Biju Parekkadan

**Affiliations:** 1Department of Biomedical Engineering, Rutgers University, Piscataway, NJ 08854, USA; srg163@scarletmail.rutgers.edu (S.R.G.); cmroth@soe.rutgers.edu (C.M.R.); 2Department of Medicine, Rutgers Biomedical Health Sciences, New Brunswick, NJ 08852, USA

**Keywords:** diffuse large B-cell lymphoma, mathematical model, clonal interaction

## Abstract

Diffuse large B-cell lymphoma (DLBCL) is one of the most common types of cancers, accounting for 37% of B-cell tumor cases globally. DLBCL is known to be a heterogeneous disease, resulting in variable clinical presentations and the development of drug resistance. One underexplored aspect of drug resistance is the evolving dynamics between parental and drug-resistant clones within the same microenvironment. In this work, the effects of interclonal interactions between two cell populations—one sensitive to treatment and the other resistant to treatment—on tumor growth behaviors were explored through a mathematical model. In vitro cultures of mixed DLBCL populations demonstrated cooperative interactions and revealed the need for modifying the model to account for complex interactions. Multiple best-fit models derived from in vitro data indicated a difference in steady-state behaviors based on therapy administrations in simulations. The model and methods may serve as a tool for understanding the behaviors of heterogeneous tumors and identifying the optimal therapeutic regimen to eliminate cancer cell populations using computer-guided simulations.

## 1. Introduction

Diffuse large B-cell lymphoma (DLBCL) is one of the most common types of lymphoid malignancies, accounting for 37% of B-cell tumors globally, with an estimated annual incidence of 15–20 per 100,000 in Europe and the USA [1]. DLBCL is known to be a heterogeneous disease, resulting in variable pathological and clinical presentations. Therefore, it has been proven necessary to modify standard cancer therapies to combat the aggressive course of DLBCL. One of the most common treatments for patients with DLBCL is the combination of the anti-CD20 antibody rituximab © alongside the small molecule drugs cyclophosphamide (C), doxorubicin (hydroxydaunorubicin, H), vincristine (Onvocin, O), and prednisone (P). Vincristine prevents mitosis by attenuating the polymerization of tubulin to form microtubules, and doxorubicin inhibits topoisomerase II in order to create DNA strand breaks. The R-CHOP regimen has considerably improved patient outcomes, resulting in disease-free rates between 50% and 70% [2]. However, this leaves 30% to 50% of people who will not be cured through this treatment, as 20% of the patients relapse after the entire treatment and 30% relapse in the middle of the regimen. About 80% of relapsed patients will die due to lymphoma, even after implementing therapies such as salvage chemotherapy or stem cell transplants [3]. It is, therefore, important to understand the mechanisms governing DLBCL development and relapse to create more effective treatment regimens and improve patient outcomes.

There have been strides to improve existing therapies to overcome therapy resistance in DLBCL. Rituximab, a chimeric IgG1 anti-CD20 antibody, has been effective at targeting CD20-positive B-cell malignancies through antibody-dependent cellular cytotoxicity (ADCC), complement-dependent cytotoxicity (CDC), and apoptosis, usually referred to as “direct cell death” (DCD) [4]. However, cells can develop resistance to the antibody through multiple mechanisms, including the downregulation or internalization of CD20, impairing ADCC through the production of C1q and reducing CDC via the consumption of complement [5]. Obinutuzumab, a second-generation Fc-engineered anti-CD20 antibody, has been developed to overcome Rituximab resistance through increased in ADCC and DCD capabilities [4,6]. The antibody is thought to bind differently to B-cells than Rituximab, decreasing its CDC potency but invoking a greater DCD response that is caspase independent [5]. Moreover, while Rituximab’s CDC is greater due to the localization of CD20 into lipid rafts, Obinutuzumab’s non-localization provides a different route to eliminate DLBCL cells. However, the effects of the antibody may be affected by the interactions between cell populations within a heterogeneous tumor.

DLBCL drug resistance is thought to be caused by multiple factors, including the cell of origin, tumor microenvironment, variabilities between patients, and clonal evolution [3,7]. A clone is a group of cancer cells that share a similar genetic profile. This clone can evolve over time through mutations and natural selection, leading to the development of a heterogeneous population of cells with diverse genetic profiles. The increased genetic diversity can lead to the acquisition of new traits, such as drug resistance, invasiveness, and immune evasion. The mixed population can also lead to clonal competition, where the mutations result in differences in growth behavior and prevalence. The clonal equilibrium, where subclone proportions are stably maintained in the tumor [8], is of interest for targeting and destroying all relevant cells within a tumor microenvironment. In clinical cases, DLBCL tumor samples contain significant heterogeneity, with clonal evolution observed in relapsed scenarios [9,10]. For example, up to seven new groups of DLBCL genetic subtypes were identified that differ from the conventional classification using immunohistochemistry (which includes germinal center B-cell–like, GCB, and non-GCB types) [11]. The continuation of cooperative functions between the two types of cells, potentially maintained via the secretion of signaling molecules, nutrients, and other diffusible factors, could result in increased drug resistance or metastatic properties. Cooperation between clones has been observed in multiple cancer types other than DLBCL. In breast cancer, tumors composed of basal and luminal genotypes can be generated via the increased secretion of the signaling molecule Wnt1, and they can evolve to rescue Wnt pathway activation in the molecule’s absence [12]. Non-genetic cooperative adaptation to therapy has been demonstrated in heterogeneous non-small-cell lung cancer tumors and has reduced sensitivity to radiation in prostate cancer [13,14].

In the present study, a mathematical model of classical predator–prey growth was explored to determine if it would predict long-term observations in lymphoma patients. We discovered that a mutualistic interaction term between parental and drug-resistant clones was essential in yielding a long-term, converging steady-state behavior. In vitro studies utilizing DLBCL cell lines that were sensitive or resistant to the antibody Obinutuzumab validated the mutualistic interaction theory, though they highlighted an imbalanced weighting of interaction between the drug sensitive and resistant cells. The mathematical model was then modified with a weighting function and further studied to see simulated events under different interaction weights. Simulations with weight-adapted growth rates were then developed using clinical growth rate parameters and under drug pressure to explore steady-state behaviors over long-term periods. These simulations are significant because they show that (1) long-term behaviors were consistent with resistant clone takeover of tumors, as well as (2) the more substantial persistence of resistant clones under clinically relevant drug pressure than otherwise known without consideration of mutualistic interactions.

## 2. Materials and Methods

Simulation Parameters

All differential equation systems were simulated using MATLAB R2020a. The “ode45” and “ode23” functions were used to compute the changes in Obinutuzumab-sensitive cell and Obinutuzumab-resistant cell populations over time [15]. The system consisted of an Obinutuzumab-sensitive cell population (*G*) and an Obinutuzumab-resistant cell population (*R*). The inputs for the system included the initial number of cells for each clone, the time (in days) of how long the simulation takes place, and a function representing the therapy concentration over time.

Initial Simulations of a Differential Equation System

A differential equation system based on the logistic growth model was used to model potential interactions that may occur between the two DLBCL clones. The Lotka–Volterra model, a system originally used to describe predator–prey population behavior, has been used to describe the interaction between prostate cancer cells [12,14]. Equations (1) and (2) include expressions needed for interactions, alongside the aforementioned logistic components. “*G*” or “sensitive cells” in the equation system represent the population consisting of DLBCL cells sensitive to therapy, and “*R*” or “resistant cells” represent the drug-resistant counterpart. The rate constants for each population and the carrying capacity are represented as “*k*” and “*N*”, respectively.
(1)$$\frac{dG}{dt}=k_{g}G\left(1-\frac{G+R}{N}\right)$$
(2)$$\frac{dR}{dt}=k_{R}R\left(1-\frac{G+R}{N}\right)$$

The logistic system was modified to incorporate theoretical interactions that may occur between the two DLBCL clones. Equations (3) and (4) include expressions needed for the interactions, alongside the aforementioned logistic components. In this case, *C_GR_* represents the interaction constant that alters the sensitive population growth rate, while *C_RG_* represents the constant modifying the resistant cell growth, and the product between both clones mirrors the extent of interactions that may occur within cultures.
(3)$$\frac{dG}{dt}=k_{g}G\left(1-\frac{G+R}{N}\right)+C_{GR}GR$$
(4)$$\frac{dR}{dt}=k_{R}R\left(1-\frac{G+R}{N}\right)+C_{RG}RG$$

Rate Constant Derivation

The rate constants were derived individually from sensitive and resistant cells in mixed cultures for each proportion (e.g., 100% sensitive cells, 100% resistant cells, 50% sensitive, and 50% resistant cells), mimicking how the separate fluorescence data can be captured in vitro. The data points until day 5 from interacting and non-interacting simulations were inputted into MATLAB’s curve fitter application (cftool) individually, and a regular logistic function was used to fit the data. The rate constant estimate from the algorithm was identified from each individual drug-sensitive or drug-resistant population and repeated over multiple proportions of each clone.

DLBCL Cell Lines, Transfection, and Transduction

SUDHL-4 (sensitive cells), a DLBCL suspension cell line, and SUDHL-4OR (resistant cells), a DLBCL suspension cell line resistant to Obinutuzumab, were used for this study. SUDHL-4 cells were procured from ATCC, and the SUDHL-4OR cell line was provided by Dr. Andrew Evens from the Rutgers Cancer Institute of New Jersey (CINJ) and derived from a patient who was resistant to Obinutuzumab and expanded ex vivo. The cells were cultured in RPMI 1640 media (ATCC 30-2001) containing 4500 mg/L glucose supplemented with 10% Gibco fetal bovine serum (FBS, 10082147) and 1% antibiotic–antimycotic solution (Fisher Scientific, Boston, MA, USA) at 37 °C in a humidified atmosphere with 5% CO_2_. Cell viability and proliferation was quantified using automated NucleoCounter NC-202^TM^ with an aliquot of 200 μL.

Cells were engineered to produce distinct fluorescent proteins in order to track them for proliferation measurements. Genetic constructs (plasmids) that constitutively produced green fluorescent protein (GFP) and red fluorescent protein (RFP) were obtained through VectorBuilder. The plasmids are as follows: pLV-EF1α-GFP and pLV-EF1α-RFP. The plasmids were packaged in lentiviral particles through transfection. HEK293T cells were cultured in Opti-MEM media with a PEI transfection reagent, a packaging plasmid (psPAX2), a viral envelope plasmid (pMD2.G), and each genetic construct in order to produce viral particles containing the constructs. A 3:2:1 ratio of the plasmid of interest, psPAX2, and pMD2.G was used. The media were replaced with RPMI 1640 24 h after the initiation of the process. The remaining media were collected 48 h after the initiation of the process and filtered using 20 mL syringes and 0.45 μm syringe filters to obtain viral particles. Concentrations of viral particles were verified via qPCR using a lentiviral titration kit acquired from Applied Biological Materials on QuantStudio 3 from Thermo Fisher Scientific (168 Third Avenue, Waltham, MA USA 02451).

In the process of spinoculation, SUDHL-4 cells were centrifuged in RPMI 1640 media containing viral particles with GFP construct, while SUDHL-4OR cells were centrifuged in RPMI 1640 media containing viral particles with RFP construct, in order to transduce the cells for fluorescent protein expression (Appendix A). The cells were centrifuged at 2400 RPM for 2 h and subsequently kept inside the centrifuge for 2 h. After culturing the transduced cells, some sensitive cells were able to express GFP, and some resistant cells were able to express RFP. The two cell populations were sorted via flow cytometry to obtain pure fluorescent cells. The process resulted in SUDHL-4 cells fluorescing green due to GFP expression and SUDHL-4 OR cells fluorescing red due to RFP expression (Appendix A).

Mixed DLBCL Cultures

Pure and mixed initial cell populations with a total initial cell number of 100,000 in 1 mL of RPMI 1640 media for each group were cultured in 24-well plates to simulate a spatial constraint. Different ratios of mixed populations were cultured in order to determine differences in rates between the cultures. Ratios of 1:0, 3:1, 1:1, 1:3, and 0:1 for G:R cells were tested. The day on which the cells were initially seeded was defined as the 0th day for the time in experimentation to match with computer models. Next, 0.5 mL media were added to each well on days 2 and 3 in order to maintain nutrition. The fluorescence and total cell count were determined each day using the Celigo imaging cytometer (Appendix A). The Celigo imaging cytometer can collect information regarding cellular fluorescence by scanning well plates without the need to remove samples from each group for flow cytometry analysis.

While scanning plates, it was noted that both cell populations tended to clump together during prolonged periods of time. The clumps of cells contained both sensitive and resistant cells, determined via the presence of green and red fluorescence, respectively (Appendix A). In order to properly image the suspended cellular clumps, the cells were separated by mixing the cultures with a pipette and subsequently centrifuged so that they lay flat on the bottom of the well plate. The in vitro data were then analyzed using MATLAB’s curve fitter application (cftool) and a conventional logistic function to fit the data and derive constants for the rate (*k*), initial population (*b*), and carrying capacity (*N*) (Appendix A).

Simulation—Differential Equation System with a Modified Interaction Expression

The system of equations (Equations (5) and (6)) included the modified interaction expression. Parameters were chosen based on the behaviors identified via in vitro cell growth. The sensitive cell growth rate constant and the resistant cell growth rate constant were derived from interpolating constants. The carrying capacity was approximated based on the total cell count that the cultures approached on day 5.
(5)$$\frac{dG}{dt}=k_{g}G\left(1-\frac{G+R}{N}\right)+C_{GR}\frac{RG^{m}}{{\left(G+R\right)}^{m}}$$
(6)$$\frac{dR}{dt}=k_{R}R\left(1-\frac{G+R}{N}\right)+C_{RG}\frac{RG^{m}}{{\left(G+R\right)}^{m}}$$

Derivation of Interaction Constants through Correlation

MATLAB’s “corrcoef” function enabled correlation with two separate sets of data. The function outputs a number close to 1 if the data sets (in vitro and simulation) are identical, 0 if they are not correlated, and -1 if they are inversely correlated. The correlation coefficients were derived through a comparison between the in vitro interaction constants and simulation interaction constants for both the sensitive cells and resistant cells. The interpolated rate constants of sensitive cells were correlated with its in vitro counterpart, and the constants of resistant cells were separately correlated with its in vitro counterpart. In order to maximize the correlation of both populations at the same time, the products of their respective correlation coefficients were assessed. If the correlation product is close to 1, it would mean that both populations would be represented accurately by one set of interaction parameters.

Simulation—Differential Equation System with Cytotoxic Antibody Drug Treatment

Equations were modified to incorporate the effects of direct cell death caused by Obinutuzumab (Equations (7) and (8)). The constants *a_G_* and *a_R_* represent the antibody’s potency against the populations of sensitive and resistant cells, respectively. The function *f(t)* represents the administration of the therapy with time as the independent variable. The function *f(t)* represents the function of therapy concentration over time. For the simulation, both potency constants were normalized so that *a_G_* would equal 1 and *a_R_* would equal 1/4.
(7)$$\frac{dG}{dt}=k_{g}G\left(1-\frac{G+R}{N}\right)+C_{GR}\frac{RG^{m}}{{\left(G+R\right)}^{m}}-a_{G}f\left(t\right)$$
(8)$$\frac{dR}{dt}=k_{R}R\left(1-\frac{G+R}{N}\right)+C_{RG}\frac{RG^{m}}{{\left(G+R\right)}^{m}}-a_{R}f\left(t\right)$$

## 3. Results

### 3.1. Tumor Clones Exhibit Different Steady-State Behaviors Based on Interactions

A mathematical model was developed that considers two cancer clones growing simultaneously, with a constraint on both populations that reflects available space and/or nutrients. A logistic function was chosen for this purpose, as it has previously been used to describe a tumor in which proliferation was constrained by the available space [16]. As both clones grow in the same location, the increase in both populations results in a decrease in both rates, as expected when space limitations restrict further growth. The total, maximum cell number reached is referred to as the carrying capacity. We first determined these growth parameters experimentally for lymphoma cell lines. Initial monocultures of DLBCL cell lines demonstrated that drug-sensitive cells grew at a faster rate than drug-resistant cells under the same experimental condition. Thus, the sensitive cell growth rate, *k_G_*, was approximated to be 1 day^−1^; the resistant cell growth rate, *k_R_* was 0.5 day^−1^; and the carrying capacity was 600,000 cells for the simulations based on those cultures.

Firstly, the growth of the two populations with a shared carrying capacity but otherwise no interactions was simulated (Equations (1) and (2)). In this case, the larger growth constant of the sensitive population resulted in more proliferation for the first 6 days (Figure 1A). Both populations ultimately reached steady states influenced by both the carrying capacity and individual growth rates—the total number of cells on day 10 approached the carrying capacity of 600,000, and the sensitive steady state was three times larger than the resistant one. A phase portrait depiction more clearly describes the distinct steady states resulting from distinct initial proportions (Figure 1B). Notably, mixed culture simulations had divergent steady-state outcomes based on the initial conditions. The mixed cultures deviated toward the sensitive cell number axis due to the population’s larger growth rate constant. The total cell number of all the cultures, including the pure and mixed variants, was equal to the carrying capacity. The divergent steady-state outcomes of mixed cultures did not intuitively correspond with biological patterns that often converge to a balanced solution or the observation of mutated cancer cell clones obtaining a survival advantage without drug pressure or an obvious genetic alteration in growth signaling.

Next, the mathematical model was modified to consider potential interactions that may occur between the two DLBCL clones, as evidenced in other cancers (Equations (3) and (4)). Parasitic interactions were simulated with one positive and one negative interaction constant (*C_GR_* = −10^−7^ cell^−1^ day^−1^ and *C_RG_* = 10^−7^ cell^−1^ day^−1^) to ensure that the resistant population had an advantage. The parasitic nature of resistant cells ultimately led to the extinction of the sensitive cell population (Appendix A). Mixed cultures with multiple ratios approached the favored population carrying capacity, regardless of the initial concentration of sensitive cells (Appendix A). Competitive interactions were also simulated with negative interaction constants (*C_GR_* = −10^−7^ cell^−1^ day^−1^ and *C_RG_* = −10^−7^ cell^−1^ day^−1^). Competitive interactions yielded two distinct steady-state solutions. The initial time response (until day 3) was similar to that of the parasitic simulation, yet mixed cultures diverged toward the pure culture steady-state solutions depending on the proportion of sensitive cells and resistant cells (Appendix A). These two interaction types, namely parasitic and competitive, did not yield converging solutions.

Mutualistic interactions, where two populations help each other during growth, were then simulated with positive interaction constants (*C_GR_* = 10^−6^ cell^−1^ day^−1^ and *C_RG_* = 10^−6^ cell^−1^ day^−1^). This depicts a scenario where both populations are equally aiding each other’s growth, for example, via the production of growth factors. As the total population neared the carrying capacity, the mutualistic interactions resulted in a different steady state—resistant cells became the dominant population, as opposed to sensitive cells, where mixed cultures always approached the steady-state solution, regardless of the initial concentration (Figure 1C). This result was consistent with a mutant clone’s ability to obtain a survival advantage without drug pressure or obvious genetic change, resulting in a faster-growing cell. This was even more apparent with the phase portrait, where mixed cultures always approached the steady-state solution, regardless of the initial concentration (Figure 1D). Interestingly, the total cell number for the steady-state solution was more than the carrying capacity due to the interaction expressions in the system. While the system can yield steady-state solutions, dramatically increasing the interaction constants may result in solutions approaching infinity. In summary, mutualistic interactions were the only simulation that had a converging solution irrespective of initial conditions or drug pressure, which allowed resistant clones to outcompete a parental cell clone.

### 3.2. Rate Constants Determine Steady-State Behaviors at Earlier Time Points

The steady-state behaviors of mixed populations that displayed mutualistic interaction were most apparent after long time periods (100 days). Yet, the experimental validation of these simulations at 100 days was intractable, as the in vitro carrying capacity was obtained quite rapidly (~5 days). There were also obvious labor and time constraints for conducting such long-term experiments. Therefore, we used an interpolation approach of growth rate constants determined from short-term culture (data until day 5) to fit into a logistic equation with or without mutualistic interaction.

We first simulated what the short-term (5 day) outcomes would look like using interacting or non-interacting models. Simulations revealed a clear difference in growth rate constants between the two models (Appendix A). In the non-interacting model, as the resistant cell percentage of mixed culture increased, the rate of sensitive cells decreased in an almost linear manner (Figure 2A). Conversely, as the resistant cell percentage decreased, the rate of resistant cells increased in a similar fashion. For a mutualistic model, the results were non-linear. Sensitive cell rates increased for mixed mutualistic cultures with small initial percentages of resistant cells, but they decreased for higher percentages (Figure 2B). Resistant cell rates in mixed mutualistic cultures demonstrated a slight decrease and a subsequent slight increase as the resistant percentage decreased. These short-term simulation results for growth rate differences between non-interacting and mutualistic models served as the basis of comparison with experimental data to determine which model better fit the actual growth dynamics.

### 3.3. In Vitro Cultures Exhibit Mutualistic Interactions between DLBCL Clones

To experimentally study an interaction model with cell-specific precision, SUDHL-4 (drug-sensitive clones) and SUDHL-4OR (drug-resistant clones) cell lines were engineered to express green fluorescent protein (GFP) and red fluorescent protein (RFP), respectively. The cell lines were then grown as pure or mixed cultures. The cultures demonstrated logistic-like growth of both pure and mixed variants with the 3:1 and 1:3 GFP:RFP ratio groups (Appendix A). The primary periods of growth were from day 1 to day 4, and then the cultures approached a steady state on day 5. The G/R ratio was computed by dividing the GFP count by the RFP count. The ratios for the mixed cultures increased until day 2, indicating an initial increase in drug-sensitive cells (Figure 3A). From day 3 to day 5, the ratio began to stabilize for the 1:3 GFP:RFP group but decreased for the other two groups. This decrease suggested an interaction in which both populations cooperated, leading to one final steady state. Similarly, in simulations with mutualistic interactions, the G/R ratio started to converge after an initial distinction between each culture, regardless of their starting proportion of sensitive and resistant cells (Figure 3B). These steady-state behaviors were independent of the initial starting populations, as was observed in original mutualistic simulations findings at long time points.

We further evaluated experimental growth rate constants as a means to validate mutualistic interactions. For the drug-sensitive cell line (GFP cells), the rate constants slightly increased as the percentage of initial resistant cells increased, with a maximum of 1.191 day^−1^ at 50% resistant cells (Figure 3C). The rate constants of mixed populations were higher than the rate of pure sensitive cell culture (1.069 day^−1^). These experimental results were consistent with a non-linear relationship of growth rate constants in mixed culture, as simulated in mutualistic interactions With regard to the drug-resistant clone (RFP cells), the rate increased, starting from the pure resistant culture (0.4695 day^−1^), as the percentage of initial resistant cells decreased, with a maximum of 0.6045 day^−1^ at 50% resistant cells. Similar to its GFP counterpart, the RFP rates of mixed cultures were higher than the pure culture. The general behavior of GFP rates was most similar to that of the rates from mutualistic interaction simulations. However, the general behavior of the RFP rates did not match any rates from the simulations—it was instead similar to the GFP cell rates observed in vitro, with an increase and a subsequent decrease as the percentage of initial resistant cells decreased. Although a mutualistic interaction was observed, a more complicated mode of interaction was likely needed to account for the disparity between in vitro results and simulations, as well as the similarity between the GFP and RFP cell rates in vitro. The model would also need to be modified to account for the steady-state cell number surpassing the carrying capacity in the initial simulations.

### 3.4. Multiple Best Fit Models Derived through Correlation

Since the first generation mutualistic model differentially captured the trends in rate constants with initial cell ratio for sensitive (GFP) vs. resistant (RFP) populations, a second-generation model was constructed that would weigh the interactions between the two cell types. This “weighting function” included a variable exponent (*m*) to the current sensitive cell number (*G*) to the conventional interaction expression and was divided by the total current cell number raised to the same exponent ([*G* + *R*]*^m^*) (Equations (5) and (6)). The exponent will determine the maximum effect of interaction based on the starting proportion of cells. For example, when *m* is equal to 1, the interaction is maximized with 50% of the initial concentration of resistant cells or via a 1:1 interaction. This form is most similar to the conventional mutualistic interaction model form. However, a maximum interaction at 50% was not what was observed experimentally. When *m* is equal to 2, the interaction was maximized with 33.3%, i.e., a 2:1 interaction ratio, of the initial concentration of resistant cells.

A weighting function was added to the interaction expressions in the differential equation system. Parameters were chosen based on the behaviors identified with in vitro cell growth. The sensitive cell growth rate constant, *k_G_* = 1.069 day^−1^, and the resistant cell growth rate constant, *k_R_* = 0.4695 day^−1^, were derived from interpolated rate constants from monocultures. The carrying capacity, *N* = 600,000, was approximated based on the total cell count that the cultures approached on day 5. Correlation tests were performed to derive interaction constants using different interaction exponents to represent in vitro data more accurately and without bias. A range of interpolated rate constants (similar to Figure 2B) were derived using a fixed exponent and a range of interaction constants *C_GR_* and *C_RG_*. The interpolated rate constants of sensitive or resistant cells were then correlated with their in vitro counterparts. The product of the independently derived correlation coefficients was maximized over the range of interaction constants. A similar method of multiplying coefficients is used in path analysis, a variation in the multiple-regression analysis used to examine causal relationships between variables [17].

Appendix A represents the correlation product when the two interaction constants are varied. The three-dimensional plot is indicative of the process used to obtain the constants—any possible combination of potential constants was checked for the correlation between the resultant simulations and in vitro data for a local maximum (Appendix A). The two-dimensional plot showed an example of how the correlation product compares to the individual correlation constants of the sensitive and resistant groups (Appendix A). Appendix A contains the different exponents used for the correlation test, along with their respective interaction constants and the correlation product. The correlation was lowest when the exponent equaled one, which was closest to the conventional mutual interaction expression. Altering the exponent yielded higher correlation products, which was a sign that our weighting approach could improve fits of data. The value of the interaction constant *C_GR_*, representing the change in interaction for the growth of sensitive cells, increased based on the exponent. Fitting constants in this manner allowed us to understand how interaction exponents affect the final steady states.

The steady-state solutions significantly varied with changes in weighted exponents (Figure 4A–D). Smaller interaction exponents resulted in the movement of steady states to the resistant cell axis, while larger exponents resulted in the dominance of sensitive cells. An exponent of 1 resulted in a similar proportion of final sensitive and resistant cells (Figure 4E). All the scenarios based on different interaction constant (*m*) values presented a possible model based on the analysis of interpolated rate constants and were carried forward into further exploratory simulations under drug selection pressure.

### 3.5. Best-Fit Models Exhibit Different Steady-State Behaviors Based on Therapy Concentration

The impacts of the Obinutuzumab therapy on the two interacting populations were tested via simulations to understand the relationship between cell growth and antibody-induced direct cell death. The direct cell death aspect of Obinutuzumab was added to the differential equation system through a time-varying function. A constant function of 1 was initially used to represent the constant presence of antibodies in the simulations. The interaction parameters with the exponent *m* = 2 were initially used, with a starting population of 50,000 sensitive and 50,000 resistant cells, alongside an end time of 1000 days. The cell numbers at day 1000 and the phase portrait demonstrated two distinct steady-state solutions (Figure 5A). At smaller concentrations of Obinutuzumab (0.3 or below), the culture approached a steady state that contained substantial amounts of both sensitive and resistant cells. The number of sensitive cells decreased as the therapy concentration increased to 0.3. This can also be seen in the phase portrait, in which distinct steady states (black dots) were achieved in the middle of the plot. However, with higher concentrations (greater than or equal to 0.31), there was an abrupt shift in the steady-state behavior (Figure 5B). The number of sensitive cells at the end became zero, and the culture was dominated only by resistant cells. Moreover, increasing the concentration continued to affect the total cell number, as the number of resistant cells continued to decrease in a steady manner. This behavior was represented in the phase portrait, with the solutions initially increasing to the right with sensitive cells, followed by a subsequent reversal in behavior as they approached the resistant cell axis (Figure 5A). These simulated results generally agree with the observation of a takeover of resistant clones under drug selection, which is eventually overcome using greater drug levels if drug resistance is assumed not to be absolute.

The critical point in steady state was a function of the cooperativity parameter, m, as well as the normalized antibody concentration. When *m* = 1, the abrupt shift did not occur. Rather, the number of sensitive cells steadily approached 0 as the antibody concentration increased until the concentration equaled 0.63 (Figure 5C,D). The resistant cell population only started to decline after the depletion of sensitive cells. When *m* = 3, the change in steady-state behaviors occurred when the concentration was 0.41. The general behavior of the end cell numbers was similar to the ones from *m* = 2, but there was a more sudden decrease in the sensitive cell number during the shift due to the large initial presence based on the interaction parameters and the exponent (Appendix A). When *m* = 0.5, the general behavior was similar to when the exponent equaled to 1, but the resistant cell population continuously decreased, regardless of concentration (Appendix A). The differences between the abrupt and non-abrupt shifts could also be identified in the phase portraits. The simulations with abrupt shifts yielded two separate clusters of final steady states. The steady-state solutions from the non-abrupt shift scenario were continuous in nature. The cluster formation seemed to be dependent on the exponent—two distinct groups formed when the exponent was greater than 1.

In summary, simulations of drug pressure tended to act in two different manners based on variations in the therapy concentrations and the exponent term in the interaction expression: (1) with exponents greater than one, there was an abrupt change in behavior from a steady state containing substantial sensitive cells to one with only resistant cells, with higher drug concentrations; (2) with exponents lower than or equal to one, there was a non-abrupt decrease in sensitive cells with higher drug concentrations.

### 3.6. Modeling Heterogeneous Tumors In Vivo with Clinical Parameters

Finally, we explored simulations the clinical impact of mutualistic interactions and therapy over periods of years. In vitro constants related to cell growth and interactions were substituted with in vivo constants determined from historical human tumor growth. The doubling time of malignant lymphoma was observed to be approximately 29 days, which translates to a tumor growth rate of 0.0239 day^−1^ [18]. We also evaluated a doubling time of 14 days as a separate fast-growing case, which resulted in a growth rate of 0.0495 day^−1^. The size of observable tumors at the time of diagnosis was assumed to be 10^9^ cells [19]. This tumor size would be the initial population of cancer cells in the simulation. Thus, a 9:1 ratio of sensitive and resistant cells, which we consider to be representative of a human tumor population, would comprise the initial population of 9 × 10^8^ sensitive cells and 10^8^ resistant cells. The carrying capacity of a tumor within the body was approximated to be 2^40^ cells, which would equal to approximately 1 × 10^12^ cells [20]. We derived a growth rate for sensitive cells by assuming a homogeneous sensitive cell population with the aforementioned initial and final cell numbers. The other constants (resistant cell growth rate constant and interaction constants) were scaled down to clinical constants based on the human sensitive cell growth values. The list of constants is presented in Appendix A.

The administration of drug also required modification to realistically model cycles of antibody treatment for patients with DLBCL. The concentration of a therapeutic decays in vivo over time as drug molecules are cleared by the body. The kinetics of a finite amount of drug administered intravenously was governed via an exponential decay function, assuming that the pharmacokinetics are described via a single-compartment pharmacokinetic model. This function took into account the initial (maximum) concentration of the antibody and the elimination constant (which modulates the speed at which the antibodies are removed from the system). In the case of a 1000 mg dose of Obinutuzumab, the maximum concentration of the antibody was 3.76 μmol/L, and its half-life was 28.4 days [21]. The elimination constant, derived using the half-life, was 0.02441 day^−1^. In a clinical trial assessing the effectiveness of Obinutuzumab, patients were given IV administrations of the antibody in 21-day cycles [22]. During cycles 2–8, patients received 1000 mg of Obinutuzumab at the start of each cycle. During the first cycle, they instead were given 1000 mg injections on day 1, day 8, and day 15 (corresponding to 0, 7, and 14 in the simulation, as the time starts at day 0). The plasma concentration of Obinutuzumab over the course of the clinical treatment with its clearance rate is shown in Appendix A, with a normalized concentration of one representing 1000 mg doses. Three initial administrations of the antibody were given to quickly increase plasma concentration, while subsequent administrations helped in maintaining a concentration range of approximately 1.5 to 2.25 times the initial dose. The concentration persisted until 400 days after the treatment was stopped on day 168. Obinutuzumab was administered alongside a regimen of CHOP that follows a similar 21-day cycle for 6 to 8 cycles. Most of the drugs present in CHOP (cyclophosphamide, doxorubicin, and vincristine) were infused via IV on day 1 of the cycle, while prednisone was given orally on days 1 to 5.

Simulations were run for 1500 days, equaling approximately 4 years, with the start of the therapy set as day 0. Without mutualistic interaction, tumors demonstrated a lack of resistant cells at lower antibody concentrations and a maximum level of them at around 0.075 (Figure 6A). When interactions were accounted for, resistant cells were found to persistent at lower antibody concentrations, though ultimately had similar end concentrations to that of the non-interacting model. In the simulation performed with slow-growing tumor constants, the behavior of total end cell numbers was consistent with both interaction exponents—tumor cells persist until the concentration reaches 0.12 (Figure 6B,C). However, the difference was more apparent when the clone populations were considered. When *m* = 1, the resistant cell population reached a peak around a concentration of 0.015 but continuously decayed until 0.12 (Figure 6C). When *m* = 2, it had two distinct maxima (when the concentration is 0.01 and 0.07), followed by a subsequent decay (Figure 6B). A large distinction can be observed between fast-growing tumors. In terms of similarity, the control and both interaction simulations resulted in the cessation of tumor cells by the 0.3 concentration. However, the interaction simulations yielded the persistent presence of the sensitive cells until certain antibody concentration thresholds, with the proportion of each clone being dependent on the exponent. When *m* = 2, the sensitive cells become depleted around the same concentration as the control simulation. Moreover, it demonstrated an abrupt shift in population proportion around that point compared to the other scenarios. When *m* = 1, however, both clones persisted for a longer concentration range until the sensitive variant gradually decreased, resulting in a peak in the resistant cell population at around 0.2. Overall, these clinical simulations under drug pressure suggest a dormant, indwelling population of resistant cells at much higher proportions when mutualistic interactions are considered. Such persistence and frequency may relate to the increased probability of secondary mutations occurring to further evade drug treatment via new mechanisms.

## 4. Discussion

In this work, a mathematical model was developed to simulate different DLBCL cells (drug sensitive vs. drug resistant) to understand the long-term behaviors of heterogeneous tumors. The model was limited in scope in order to study potential interactions between two cancer populations. Multiple assumptions were used to describe initial cell growth and drug resistance. A logistic function was chosen to describe the general behavior of DLBCL cells, as it can be relevant when considering a spatially extended system in which proliferation is constrained by available space [16]. Moreover, logistic growth has been observed in clonal equilibrium associated with chronic lymphocytic leukemia, another hematological malignancy [23]. For initial computational simulations, it was assumed that the resistant cells would have a smaller rate constant than the sensitive variant. Drug-resistant cancer cells have been observed to grow at a slower rate, suggesting that reduced proliferating activity can contribute to resistance [24,25].

We discovered that only models that accounted for mutualistic interaction could lead to a converging, steady-state solution that was independent of starting conditions after long periods of time. Experimental validation was then performed to compare to simulated, interpolated rate constants that occurred at smaller timescales. Cells sensitive and resistant to therapy were engineered to produce fluorescent markers and cultured as mixed populations to determine which model better fit the experimental data. A mutualistic interaction model showed more agreement with experimental data, though the two cell types showed unique patterns, which led us to further modify the interaction expression. The fit was improved in a cell-type manner with higher values of cooperativity, *m*, supporting the introduction of this parameter into the model. As the improvement was moderate, it was decided to continue with all four values of *m* for the further analysis of scenarios involving therapeutic treatment to see if one was further aligned with experimental observations.

Interactions between tumor cells and the surrounding non-tumor cells play an important role in promoting cancer cell proliferation or metastasis. In in vitro studies, mixed cultures tended to form free-floating clump-like structures, which contained both sensitive (green) and resistant (red) cells. The process of clumping could be a potential mechanism through which interaction between the two types of cells occurs. In this work, we incorporated a non-capacity-constrained interaction term (Equations (3) and (4)) to account for the possibility that the capacity of both space and nutrients is likely to be altered by interacting cell types. Nonetheless, a spatial component considering the distance between two cancer clones (such as a partial differential equation system) could more accurately depict their interactions within the spheroid-like structures that were observed here. Partial differential equations can be used in that case to understand the spatial variations that occur during tumor proliferation [26].

Biologically, multiple mechanisms could be involved in conferring advantages for tumor cells in heterogeneous clusters, and they are worthy of follow-up experimentation. Tumors could produce growth factors, which can bind to their own surface or the surface of genetically similar clones, leading to enhanced proliferation due to autocrine signaling. The use of genetic or epigenetic analyses and growth factors in cultures can aid in determining which biochemical pathways the cancer clones use for interactions. For example, the methylation of TGF-β associated genes was linked to therapy relapse in DLBCL cells [27]. Thus, the use of TGF-β in pure and mixed cultures could reveal the alteration of growth patterns indicative of a change in interactions. The overexpression of epidermal growth factor receptor (EGFR) is known to enhance cell survival in epithelial cancers and gliomas [28,29]. Moreover, cancer stem cells (CSC), a variant with stem cell-like properties, can interact with their non-CSC counterparts to enhance survival. Differentiated colorectal cancer cells have been known to protect the stem-cell variants from chemotherapeutic toxicity, and glioma cells behaving like CSCs can produce IL-6 to promote the growth of non-CSCs [30,31].

To incorporate these types of interactions into simulations, a model would need the inclusion of the genetically similar cell population with information regarding its growth and self-renewal capacity. Compartment models describing the movements of cells between different types and competition models describing interactions between different types have been used to model the development of therapy resistance and heterogeneous tumors, respectively [14,32]. Moreover, it could involve tracking the molecular kinetics of signaling molecules and receptor-binding processes. Paracrine signaling from other non-cancerous cells in the tumor microenvironment has also been known to enhance tumor growth. Stromal fibroblasts can promote tumor progression through remodeling the extracellular matrix and producing cytokines or transforming growth factor-β (TGF-β) [33,34]. Cancer cells can also take advantage of immunomodulatory activities in the microenvironment to promote growth and metastasis. Cells expressing PD-L1 checkpoint can interact with PD-1 on T cells, leading to exhaustion and decreased antitumor responses [35]. Tumor-associated macrophages (TAMs) can lean toward the M2 phenotype and aid tumor cells by producing immunosuppressive cytokines and other factors to promote angiogenesis or chemotherapy resistance [33,36,37]. Macrophages can additionally influence phenotypic changes in cancer stem CSCs, leading to increased metastatic potential [38]. The inclusion of TAMs or effector cells would require the tracking of the non-cancerous cell populations, which may require greater incorporation of in vivo data, as they are not self-renewing in the same capacity as cancer cells. Systems of ordinary differential equations consisting of tumor cells, antibody concentrations, and effector cells that attack cancer may better reflect the in vivo dynamics of therapies [18,39]. 

The effects of drug treatment Obinutuzumab were also simulated under mutualistic interactions, revealing a dormant population of drug-resistant cells. While Obinutuzumab has multiple modes of action to induce apoptosis, ADCC requires the presence of effector immune cells to kill CD20-positive cells. Thus, for simplifying the current model, only the effects of direct cell death were analyzed instead of adding another cell population (effector cells). For the therapeutic effects of the antibody, it was assumed that the effect of the therapy is only dependent on the concentration of the molecules in the *f(t)* expression and not dependent on the effects of ligand-binding conformation changes that are characterized by the Hill equation. Assumptions regarding the potency were also made to simulate therapy effects. In one study, high concentrations of Obinutuzumab alone resulted in a proliferation percentage of 75%, starting from a percentage of 100% without the antibody for an Obinutuzumab-resistant SUDHL-4 clone [6]. Assuming that the proliferation percentage for Obinutuzumab-sensitive cells would be 0% with a high antibody concentration, it can be interpreted that the therapy is one-fourth as effective against the resistant counterparts. Implementing other forms of therapy requires using different mathematical expressions to depict their effects. The linear quadratic model incorporating tissue sensitivity constants is used to describe the effects of radiation therapy based on similar behaviors observed in kill curves [40]. Chemotherapy modeling, on the other hand, uses a similar expression to the one used in this work to describe antibody effects, in which the dose and population number decreased the growth rate [41]. As both of the aforementioned treatment modalities are less specific, they would likely require less information on in vivo and microenvironment conditions to accurately predict responses. Since antibody therapy is more targeted, a more complex model would incorporate the effects of direct cell death with receptor binding kinetics and the effector cell populations (such as T cells or NK cells) with their respective affinities to the molecule for the effects of ADCC. 

Upon treatment with a drug that induces direct cell death, two different behaviors were identified in terms of the population steady states, which depend on the interaction exponents. Some interaction exponents resulted in two distinct steady states, while others resulted in a singular steady state, as the therapy concentration increased. When the exponent was less than or equal to 1, the steady-state cell numbers for sensitive and resistant cells gradually decreased with higher therapy concentrations, resulting in a spectrum of steady states, as observed in the phase portraits. However, exponents greater than 1 yielded steady states with substantial proportions of sensitive and resistant cells with lower therapy concentrations, but abruptly shift to pure resistant cultures after surpassing a threshold drug concentration. Moreover, testing the model with in vivo parameters revealed variations in resistant cell population behaviors in slow-growing and fast-growing tumor cases. In terms of slow-growing tumors, both interaction types had similar tumor profiles throughout the concentration range, in which there was a mild persistence of resistant cells. In fast-growing tumors, the simulations demonstrated distinct behaviors based on interaction types. When *m* = 2, both sensitive and resistant cells persist in an equilibrium until an abrupt shift occurs at the 0.99 concentration, which results in the depletion of green (sensitive) cells and an increase in red (resistant) cells. In contrast, the simulation with no interaction only has sensitive cells at earlier concentrations, and the resistant cells become the dominant clone in a more gradual manner. When *m* = 1, however, the resistant cells are dominant throughout the concentration range and approach a maximum after the eventual depletion of sensitive cells. It is, therefore, likely that a mutualistic interaction exists between drug-resistant and drug-sensitive cells based on a converging, long-term steady-state solution and the dormant persistence of resistant cells under drug pressure. These simulated findings better relate to actual clinical observations.

The results of human-based studies in the literature do not completely suggest the validity of a model with one interaction exponent over another. Cooperativity, as defined by our exponential term *m,* has been seen in other cancers and its impact on drug treatment. A clinical ratio of cancer to stroma cells was used to predict the invasiveness of lung adenocarcinoma and head and neck cancers [42,43]. Effector-to-target cell ratios are also important for determining the effectiveness of the immune system against tumors, with a ratio of 100:1 maximizing effectiveness for neutrophil ADCC [44,45]. A primary method of studying the variations between regular and relapsed/refractory DLBCL is through the mutational analyses of tumor biopsies. These studies have demonstrated that specific gene alterations necessary for immune surveillance and suppression are observed at higher frequencies in samples after relapse, and some of them are correlated with reduced overall survival [6,46,47]. The patients would have likely been treated with therapies based on a constant dose or body weight, and analyses after relapse do not capture the dynamics involved between therapy-sensitive and therapy-resistant clones. Thus, an in vivo study analyzing tumors at multiple time points, from a diagnostic size to the time of relapse (potentially by tracking general tumor growth behaviors via growth rate constant derivation), would give a better understanding of which best-fit model would most efficiently capture real-life observations. For example, an abrupt shift in clonal populations because of increasing therapy concentration would favor larger interaction constants. The extent of cooperativity in our model simulations remains indeterminate without further validation, though we speculate that it is likely less than or equal to 1 given the more plausible gradual, non-abrupt switching of sensitive to resistant clonal evolution.

In conclusion, this mutualistic interaction model could be useful as a clinical predictor of heterogeneous tumor growth and aid in optimizing therapy regimens. The results indicate a need to evaluate interclonal interactions and use different approaches to reduce specific cell populations. While conventional therapy may be more necessary for clearing drug-sensitive cancer cells in the cases where the interaction exponent equals one, altering it to a different medication to coincide with the abrupt shift of cell populations noted with larger exponents could be more efficient at reducing the tumor burden. Both the patients’ tumor growth datasets and mutation profiles can be used to assess the interaction exponents involved and predict future proliferation behavior. Therapy concentration and duration can subsequently be altered to be most effective during different phases of tumor growth. Combining predictive mathematical models with specific regimens enables physicians to offer personalized medicine to treat patients more effectively.

## Figures and Tables

**Figure 1 bioengineering-10-01360-f001:**
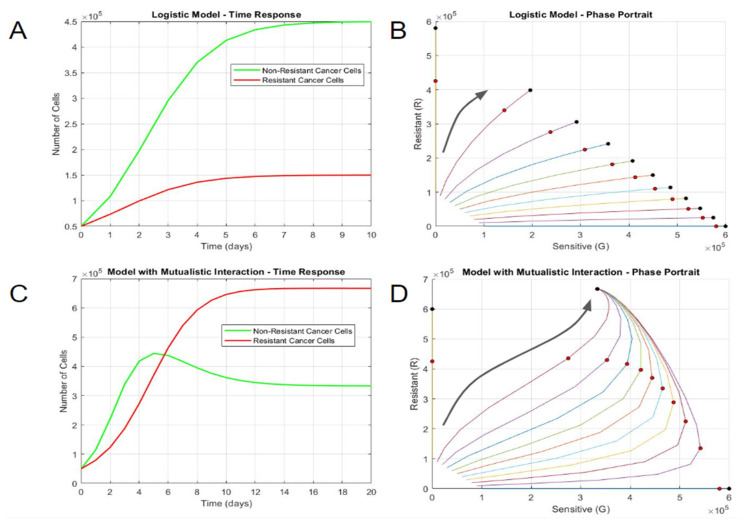
Long-term growth behaviors differentiate non-interacting and interacting cell populations. (**A**) Time response and (**B**) phase portrait of logistic model without interaction. (**C**) Time response and (**D**) phase portrait of logistic model with mutualistic interaction. In the phase portraits, the horizontal axis represents the number of sensitive DLBCL cells, and the vertical axis represents the number of resistant DLBCL cells. Each curve in the phase portrait was created by altering the percentage of sensitive and resistant cells while keeping a constant initial population. The larger black arrows represent the general trajectories of population growth. The red circles represent the states on day 5, while the black circle represents the steady state(s) at much longer time periods (day 100).

**Figure 2 bioengineering-10-01360-f002:**
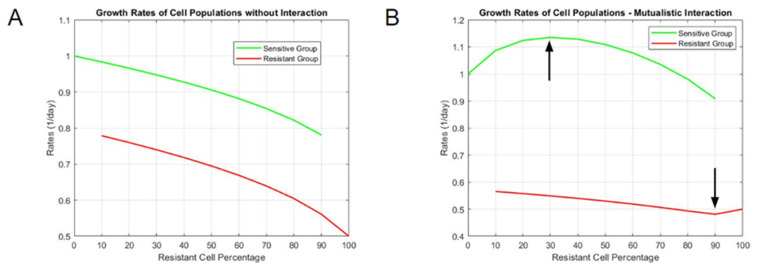
Interpolated growth rate constants of mixed cultures (**A**) without interaction and (**B**) with mutualistic interaction. The vertical axis represents the magnitude of rate constants, and the horizontal axis represents the resistant cell percentage in each tested population proportion (a resistant cell percentage of 60 means that the rate constants for each clone are derived from a starting culture of 40% sensitive DLBCL cells and 60% resistant DLBCL cells).

**Figure 3 bioengineering-10-01360-f003:**
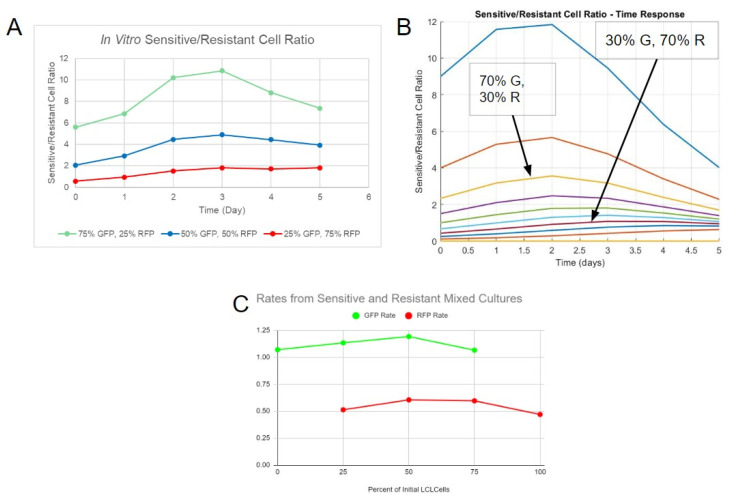
Data from in vitro study indicates mutualistic interactions between DLBCL clones. (**A**) Ratios of sensitive/resistant cells of three tested proportions from in vitro fluorescence data. (**B**) Ratios of sensitive/resistant cells of multiple proportions over time from simulation. (**C**) Interpolated growth rate constants derived from fitting fluorescence data with the standard logistic function.

**Figure 4 bioengineering-10-01360-f004:**
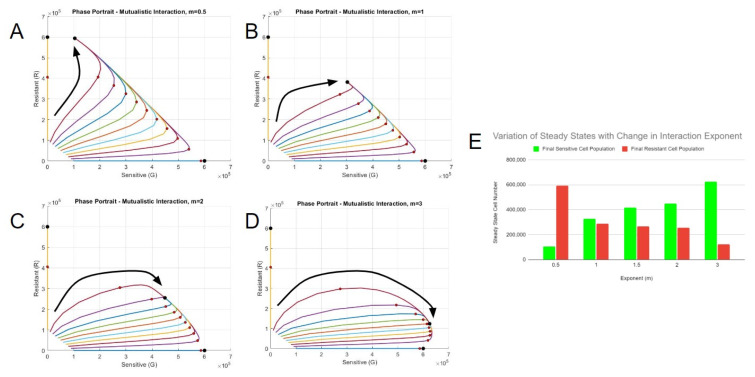
Correlation yields multiple best-fit models. Phase portraits of systems with best-fit constants with the interaction exponents (**A**) *m* = 0.5, (**B**) *m* = 1, (**C**) *m* = 2, and (**D**) *m* = 3. The red circles represent the states on day 5, and the black circle represents the steady state(s). The larger black arrows represent the general trajectories of each scenario. (**E**) The final number of cells at steady state for each interaction exponent scenario.

**Figure 5 bioengineering-10-01360-f005:**
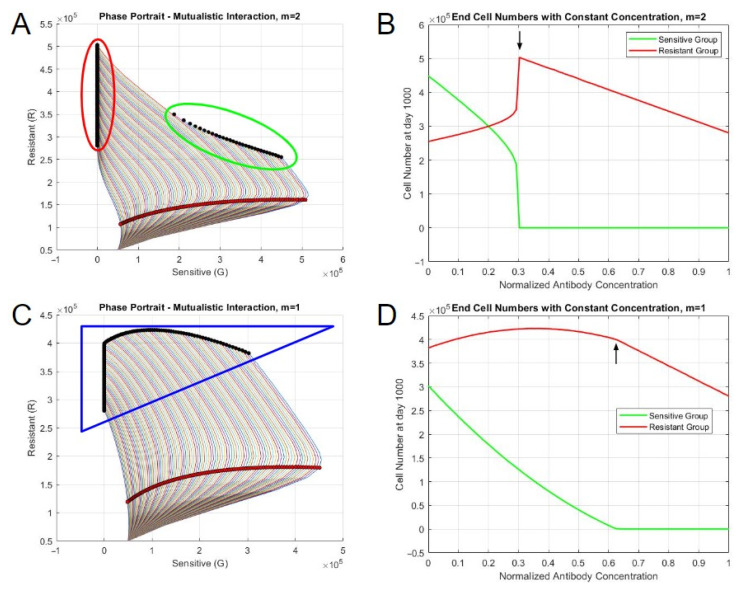
Steady-state behaviors of systems with constant therapy. (**A**) Phase portrait when *m* = 2. (**B**) End cell numbers when *m* = 2. (**C**) Phase portrait when *m* = 1. (**D**) End cell numbers when *m* = 1. The red shaded circles represent the states on day 5, and the black shaded circles represent the steady states. The outlines encircling sets of steady states represent distinct behaviors based on the concentration of therapy.

**Figure 6 bioengineering-10-01360-f006:**
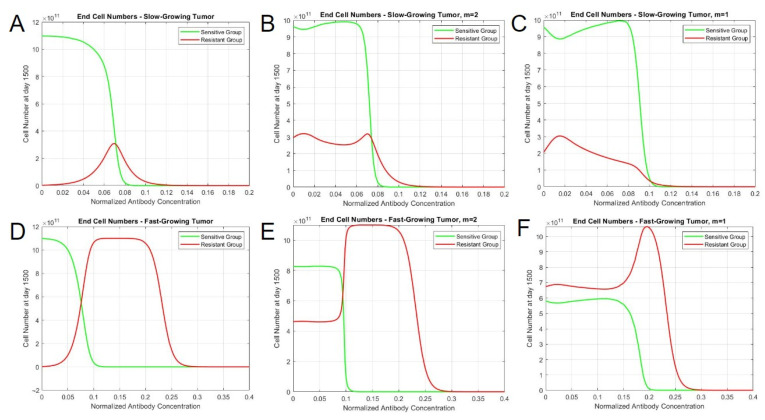
Tumor composition varies based on interaction type, tumor growth, and therapy concentration. End cell numbers for (**A**) slow-growing tumor, without interaction; (**B**) slow-growing tumor, *m* = 2; (**C**) slow-growing tumor, *m* = 1; (**D**) fast-growing tumor, without interaction; (**E**) fast-growing tumor, *m* = 2; and (**F**) fast-growing tumor, *m* = 1. The vertical axis represents the cell number at day 1500, and the horizontal axis represents the magnitude of the antibody concentration with a constant treatment cycle.

## Data Availability

The data presented in this study are available upon request from the corresponding author. The data are not publicly available due to privacy.

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
