# Peer review of "Simulating Interclonal Interactions in Diffuse Large B-Cell Lymphoma"

_bioengineering, 2023, doi:10.3390/bioengineering10121360_

Round 1
Reviewer 1 Report
Comments and Suggestions for Authors
This research made in silico computational modeling, and in vitro simulations of the grow of DLBCL lymphoma cells lines. It is a very specialized study, with single cell population and mixed, sensitive and resistant to anti-CD20. The manuscript is well written, it is "easy" to read, and to understand.
Comments:
(1) Please delete lines 23-29 as they are the mdpi template.
(2) In the Introduction, regarding the heterogeneous disease statement (line 34). Could you please add (very briefly) some recent advances in the classification of DLBCL that highlights the heterogeneous situation? For example, please look into https://doi.org/10.1182/blood.2022015851 and cite accordingly. Also, you may refer to WHO 4th or 5h editions.
(3) Line 37. Could you please explain the mechanism of action or rituximab and the other components of CHOP regimen on the B-lymphocytes. There are several hypothesis\evidences\mechanisms. (Later appear in line 384, but in introduction may also be useful).
(4) Line 82 and 83. Please add sensitive or resistant to Obinutuzumab.
(5) Please add the properties and mechanism of action of Obinutuzumab
(6) Line 81. Regarding the functions. You may need to cite
[1] Dormand, J. R. and P. J. Prince, “A family of embedded Runge-Kutta formulae,” J. Comp. Appl. Math., Vol. 6, 1980, pp. 19?26.
[2] Shampine, L. F. and M. W. Reichelt, “The MATLAB ODE Suite,” SIAM Journal on Scientific Computing, Vol. 18, 1997, pp. 1?22.
(7) Line 120. Could you please expand the descripton of the two cells lines? The SUDHL-4 is commercially available, but not sure about the SUDHL-4OR. How was the sensitivity and resistance to the anti-CD20 tested?
(8) Where possible, please add the company and catalog numbers of all reagents.
(9) Lines 148-149. Regarding "The process resulted in SUDHL-4 cells fluorescing green due to GFP expression and SUDHL-4 OR cells fluorescing red due to RFP expression". Could you please make a figure showing the engineering of producing the green and red fluorescence in proliferating cells?
(10) Are all the mathematical formulas from Matlab, or some were of own design? Please confirm that there are not mistakes in them.
(11) Regarding lines 128-129. "Initial monocultures of DLBCL cell lines demonstrated that drug-resistant cells grew at a faster rate than drug-sensitive cells". In Figure 1A, the logistic model shows that proliferation is lower in the resistant cancer cells; it reaches the plateau (steady state) faster and has "less slope".
(12) Results 3.1 and 3.2 show the modeling. However, DLBCL is not only comprised of B-lymphocytes, but also has an important component of the immune microenvironment. Does the model includes microenvironmental factors?
(13) Line 300. When making mixed cultures, how do you ensure the same number of cells are present at the startpoint?
(14) Lin Figure 3.A. Do the lines of different color represent 3 types of mixed cultures? I understand that the data doesn't have dispersion data? At days 2-3 there is an increase of the sensitive/resistance cell ratio. Is it due to increase/decrease of sensitive/resistant cells?
(15) Could this method be applied to circulating DLBCL cells from patients, to test the biological properties of the lymphoma, and the sensitivity to drugs (anti-CD20)?
(16) Does the computational modeling agree with the cell line in vitro behavior all the times? Which computational modeling fitted better the in vivo data?
Reviewer 2 Report
Comments and Suggestions for Authors
The subject is of interest and the approach could be too. The introduction is excellent and the description of the methodologies correct.
A general problem during reading appears. The establishment of the first simulations is not argued, why these parameters and especially these values. The manipulations concern 2 separate populations (susceptible or non-susceptible). And, then, it is not easy to follow the link between experiments and simulations. This is critical with the most complex models. It seems that it is more research on the influence of parameters on equations than models having the possibility of (i) reproducing complex biological data and (ii) being usable.
In addition, often long texts make monitoring difficult and do not clearly convey the strength of the approach.
Reviewer 3 Report
Comments and Suggestions for Authors
This is a very interesting paper dealing with the mathematical modelling of the interactions between sensitive and resistant Diffuse large B-cell lymphoma cells, to investigate the evolving dynamics between parental and drug-resistant clones with the same microenvironment.
The topic is worthy of investigation and well fits with the scope of the journal.
The experimental set-up is well designed and performed and is suitable for the obtainment and validation of the results, which agree with the authors hypothesis.
The presentation of the results is of high quality and a proper discussion is inserted to underline the impact of the study among the scientific community.
It is an opinion of this reviewer that the paper can be published in its current form without significant changes.
Round 2
Reviewer 2 Report
Comments and Suggestions for Authors
Authors have provided some pertinent sentences to follow their study. It is simple but change a lot the reading, and diminish the doubt reader can have.